# Genetic relationship and source species identification of 58 Qi-Nan germplasms of *Aquilaria* species in China that easily form agarwood

Yong Kang[1], Peiwei Liu[1], Feifei Lv[1], Yuxiu Zhang[1], Yun Yang[1]*, Jianhe Wei[1,2]*

1 Hainan Provincial Key Laboratory of Resources Conservation and Development of Southern Medicine & Key Laboratory of State Administration of Traditional Chinese Medicine for Agarwood Sustainable Utilization, Hainan Branch of the Institute of Medicinal Plant Development, Chinese Academy of Medical Sciences and Peking Union Medical College, Haikou, China, 2 Key Laboratory of Bioactive Substances and Resources Utilization of Chinese Herbal Medicine, Ministry of Education & National Engineering Laboratory for Breeding of Endangered Medicinal Materials, Institute of Medicinal Plant Development, Chinese Academy of Medical Sciences and Peking Union Medical College, Beijing, China

* yangyun43@aliyun.com (YY); wjianh@263.net (JW)

**Data Availability Statement:** All accession numbers are available from the NCBI GenBank database (https://www.ncbi.nlm.nih.gov), and the

## Abstract

Recently, Qi-Nan germplasm, the germplasm of *Aquilaria* species that easily forms agarwood, has been widely cultivated in Guangdong and Hainan Provinces in China. Since the morphological characteristics of Qi-Nan germplasm are similar to those of *Aquilaria* species and germplasm is bred by grafting, it is difficult to determine the source species of this germplasm by traditional taxonomic characteristics. In this study, we performed a DNA barcoding analysis of 58 major Qi-Nan germplasms as well as *Aquilaria sinensis*, *A. yunnanensis*, *A. crassna*, *A. malaccensis* and *A. hirta* with 5 primers (nuclear gene internal transcribed spacer 2 (ITS2) and the chloroplast genes *matK*, *trnH-psbA*, *rbcL* and *trnL-trnF*). This field survey in the Qi-Nan germplasm plantations in Guangdong and Hainan Provinces aimed to accurately identify the source species of Qi-Nan germplasm. According to the results, ITS2 and *matK* showed the most variability and the highest divergence at all genetic distances. This ITS2+*matK* combination, screened for with TaxonDNA analysis, showed the highest success rate in species identification of the Qi-Nan germplasm. Clustering in the phylogenetic trees constructed with Bayesian inference and maximum likelihood indicated that the Qi-Nan germplasm was most closely related to *A. sinensis* and more distantly related to *A. yunnanensis*, *A. crassna*, *A. malaccensis* and *A. hirta*. Therefore, this study determined that the source species of the Qi-Nan germplasm is *A. sinensis*.

## Introduction

Agarwood is resinous wood produced when *Aquilaria* or *Gyrinops* species (of the Thymelaeaceae family) are injured [1]. This substance is a valuable natural perfume and is used in traditional Chinese medicine to relieve pain and warm the middle to reduce vomiting [2].

reference numbers [OM908943-OM909007, OM938993-OM939187].

**Funding:** This work was supported by National Key Research and Development Program of China (2018YFC1706400), General project of Hainan Provincial Natural Science Foundation of China (322MS144), CAMS Innovation Fund for Medical Sciences (2021-I2M-1-032) and Advanced talents project of Hainan Provincial Natural Science Foundation of China (321RC661). The funders had no role in study design, data collection and analysis, decision to publish, or preparation of the manuscript.

**Competing interests:** The authors have declared that no competing interests exist.

Worldwide, agarwood has widely featured in cultural, religious, and medicinal practices as well as other areas [3]. In the past, *Aquilaria* species were identified solely on the morphological characteristics of the flowers, seeds and fruits. However, this identification method can be subjective, so the classification of *Aquilaria* species remains unclear. Most studies have suggested that there are approximately 20 *Aquilaria* species in the tropical regions of Southeast Asia [4, 5], but in China, only *A. sinensis* and *A. yunnanensis* have been recorded [6]. Because of agarwood's high economic, collection and medicinal value, it is increasingly sought worldwide, which has led to the damage and destruction of *Aquilaria* species. In addition, wild *Aquilaria* resources have increasingly been exhausted due to urbanization, especially in India, Myanmar, Malaysia, Vietnam, Indonesia and other Southeast Asian countries. Therefore, in order to minimize the harvesting and destruction of wild *Aquilaria* species, about 20 *Aquilaria* species have been listed in the International Union for the conservation of nature (IUCN) Red List of Threatened Species in 1998 [7], and have also been listed in Appendix II of the Convention on International Trade in Endangered Species (CITES) in 2005 [8].

Currently, one of the most effective ways to solve the shortage of agarwood is to cultivate *Aquilaria* species. China was one of the first countries to alleviate the loss of wild agarwood and protect wild agarwood resources by cultivating *Aquilaria* species, mainly through seed propagation, tissue culture, cutting and grafting propagation [9]. Since the 1980s, large-scale seed reproduction of *Aquilaria* species has taken place in Hainan, Guangdong, Guangxi, and Yunnan Provinces as well as other places in China [10]. However, most of the seeds used for seed propagation have originated from previously cultivated or wild mature plants. Seeds of *Aquilaria* species cannot be stored for long periods, and this, coupled with the lack of systematic breeding and germplasm confusion, has led to poor agarwood quality and yields [11]. In tissue culture of *Aquilaria* species, rooting is the main factor restricting reproduction [12]. Cutting propagation of these species is difficult to control, and plants have low survival rates according to Niu et al., the survival rate of lignified cuttings was 22% in summer but only 12% in winter [13]. Finally, the beneficial traits of the parent *Aquilaria* species can be preserved through grafting propagation [14].

Germplasm resources carry genetic information and have actual or potential utilization value, they mainly include material from plants, animals, and microorganisms. Plant germplasm resources mainly include those for crops, traditional Chinese medicine and forests [15]. Qi-Nan germplasm is a forest germplasm resource resulting from grafting propagation and retains the excellent agarwood-forming predisposition of its parents. Grafting propagation can also be used to obtain germplasm resources that are easy to collect and genetically stable, thereby protecting wild *Aquilaria* species. In recent years, farmers have relied on experience to find wild, highly fragrant *Aquilaria* species in Dianbai, Guangdong Province. They then transplant these species to their homes to serve as a Qi-Nan germplasm seed tree and use the branches of the seed tree as the scion for grafting propagation. However, the main propagation method of farmers is grafting the branches of Qi-Nan germplasm seed trees to cultivated *Aquilaria* species. Most of these seed trees come from Huizhou, Maoming, Shenzhen, Hong Kong, Hainan Province and other places in China. Thus, Qi-Nan germplasm easily forms agarwood, and the yield and extract content of its agarwood are higher than those of general agarwood.

In recent years, Qi-Nan germplasm has been extensively cultivated in Guangdong, Guangxi, and Hainan Provinces as well as other places in China due to agarwood's scarcity and value. Each grower claims that the agarwood produced by their Qi-Nan germplasm has a high oil content, strong fragrance, and is rapidly formed. However, the source species of many Qi-Nan germplasms remain unclear. The source species has variously been proposed to be a domestic *Aquilaria* species, an alien *Aquilaria* species, or even a new species. At present, many Qi-Nan germplasms are cultivated in China, with substantial variability in plant size, leaf

shape, stem morphology and agarwood-forming performance. However, the source species and genetic relationship of these germplasms remain unknown, which limits their use and protection. According to this review, the source species of different Qi-Nan germplasms and their genetic relationships are the main factors restricting the application and promotion of this germplasm.

DNA barcoding has been widely applied to identify *Aquilaria* species. Jiao et al. extracted DNA from *A. sinensis* wood tissue and reported that *trnL-trnF* and ITS1 could be used to construct a phylogenetic tree of *A. sinensis* [16]. Lee et al. found that the phylogenetic tree constructed by with *trnL-trnF* and ITS2 could be used to identify *Aquilaria* species [5]. Additionally, Li et al. analyzed *A. sinensis*, *A. yunnanensis* and *A. crassna* with three DNA barcodes and found two combinations (ITS+*matK* and ITS+*trnL-trnF*) that could identify these three species [17]. Eurlings et al. further suggested that the *trnL-trnF* fragment provides a new method of molecular identification for *Aquilaria* and *Gyrinops* species [18]. In a preliminary study of *Aquilaria* species from different countries, *matK* and combinations thereof facilitated the accurate identification of multiple *Aquilaria* species [19]. In summary, although DNA barcoding allows the identification of many *Aquilaria* species, the application of DNA barcoding fragments or combinations of materials differ substantially. In addition, few reports have utilized DNA barcoding to identify the main Qi-Nan germplasm. In this study, DNA barcode technology was used to determine the molecular identification of Qi-Nan germplasm and in subsequent analysis to explore differences in the applicable fragments or combinations.

Thus, we used DNA barcode technology to identify the source species of Qi-Nan germplasm. In this study, we selected 58 different types of Qi-Nan germplasms from popular markets and included *A. sinensis*, *A. yunnanensis*, *A. malaccensis*, *A. crassna*, and *A. hirta* as the research objects. Five DNA barcode sequences (ITS2, *matK*, *trnH-psbA*, *rbcL* and *trnL-trnF*) were compared in sequence to screen for the barcode fragment or combination most suitable for identifying the source species of Qi-Nan germplasm. Then, the phylogenetic trees of the Qi-Nan germplasm and the five *Aquilaria* species were constructed with the best combination. Finally, the source species of the Qi-Nan germplasm was identified according to the clustering of the phylogenetic trees.

## Materials and methods

### Materials

A total of 65 test materials were used in this study (S1 Table). The Qi-Nan germplasm plantation in Guangdong and Hainan Provinces provided 58 mainstream Qi-Nan germplasm resources. Their Chinese folk names are as follows: Qianlixiang, Jinshaozi, Lvyouwang, Youyezi, Wuming2, Jinshaye, Jianyezi, Ziqi, Zhangshaozi, Genmaidaye, Dayepo, Lanbaoshi, Putaoteng, Honggujianye, Wuming1, Tianxiang2, Tieshao, Xiaoyezi, Jianyewang, Wushenyuanye, Toudinglv, Xiguaye, Kuaishu, Zhitianjiao, XiangFei, Jianyeliu, Tangjie, Youbawang, Xiangjian, Xiangyouwang, Tianxiang1, Heizhenzhu, Jinlv, Ruhuchangye, Ruhuyuanye, Youshao, Zidantou, Dashao, Xiangshi3, Xiangshi4, Xiangshi8, Xiangshi13, Zhouyezi, Ziluolan, Diwanggu, Shisansuo, Yuanyobolang, Bohelv, Liuyouwang, Yinggelv, Diaoyouwang, Xiaoyingtao, Baozhayou, Hutoumen, Huizhouchenxiang, Zhongshannizhong, Guanxiang1, and Guanxiang2. In addition, we also selected 5 *Aquilaria* species that were accurately identified in the previous stage for comparison: *A. sinensis*, *A. yunnanensis*, *A. malaccensis*, *A. crassna* and *A. hirta*. Fresh and intact leaves of the 65 samples were collected and dried with silica gel for preservation. The grafting and agarwood-forming process as well as the whole Qi-Nan germplasm plant are shown in Fig 1 and S1 Fig. The fruits of the *Aquilaria* species used in this study are shown in Fig 2.

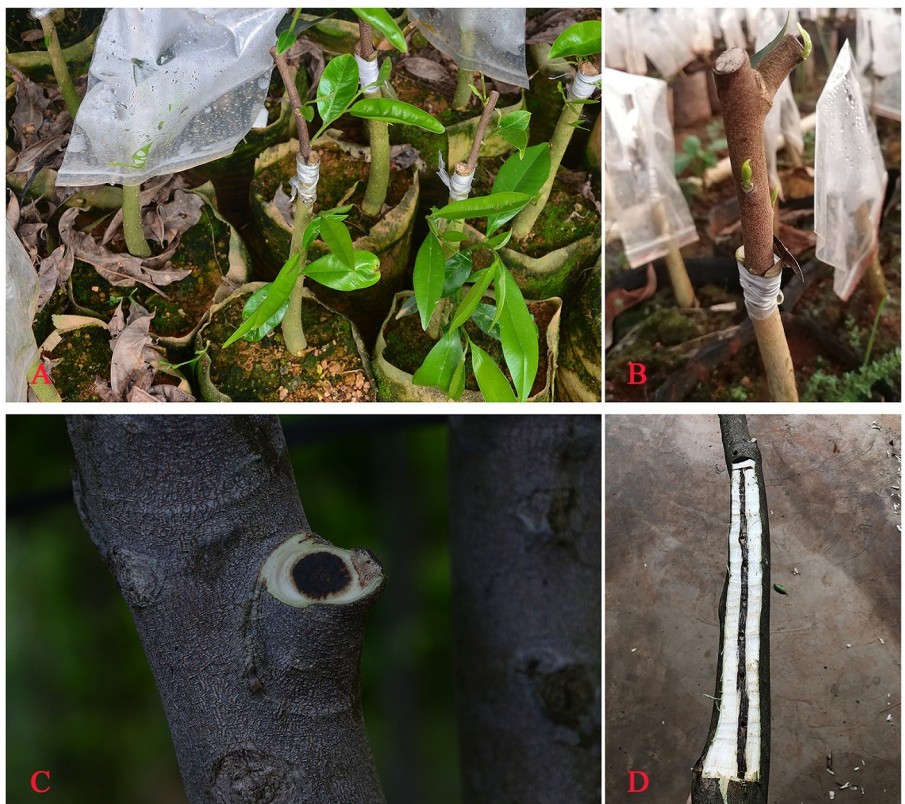

**Fig 1. Grafting and agarwood formation with Qi-Nan germplasm.** A and B: Grafting of Qi-Nan germplasm. C and D: Agarwood formation with Qi-Nan germplasm.

## Methods

**DNA extraction, PCR amplification and sequencing.** Genomic DNA of the 65 samples was extracted using the Plant Genomic DNA Kit (Tiangen Biotech, Beijing, China), following the manufacturer's instructions. Universal primers were used for PCR amplification of ITS2, *matK*, *trnH-psbA*, *rbcL* and *trnL-trnF*. Final volume of each PCR reaction was 25ul and contained 2ul DNA template, 1ul each of forward and reverse primers, 12.5ul PCR Master Mix (2X, Tiangen Biotech, Beijing, China) and 8.5ul deionized water. Optimization and adjustments were made according to the PCR conditions reported in Table 1 [20]. The PCR products were resolved by electrophoresis on 1.5% agarose gel. All amplified products were sequenced by Guangzhou IGE Biotechnology Co., Ltd. The DNA sequencing method used the Sanger method, and DNA sequencing was performed in ABI PRISM 3730xl Genetic Analyzer (Applied Biosystems, USA).

## Data analysis

Sequence editing, alignment and splicing, and computation of genetic distances were mostly completed in BioEdit [26] and Sequencematrix [27]. The PCR amplification success rate and sequencing success rate were determined following Kress [28]. Information on the length of amplification, variable sites, conserved sites, parsimony informative sites, singleton sites and genetic distances of each fragment was collected in MEGA X [29]. The species identification

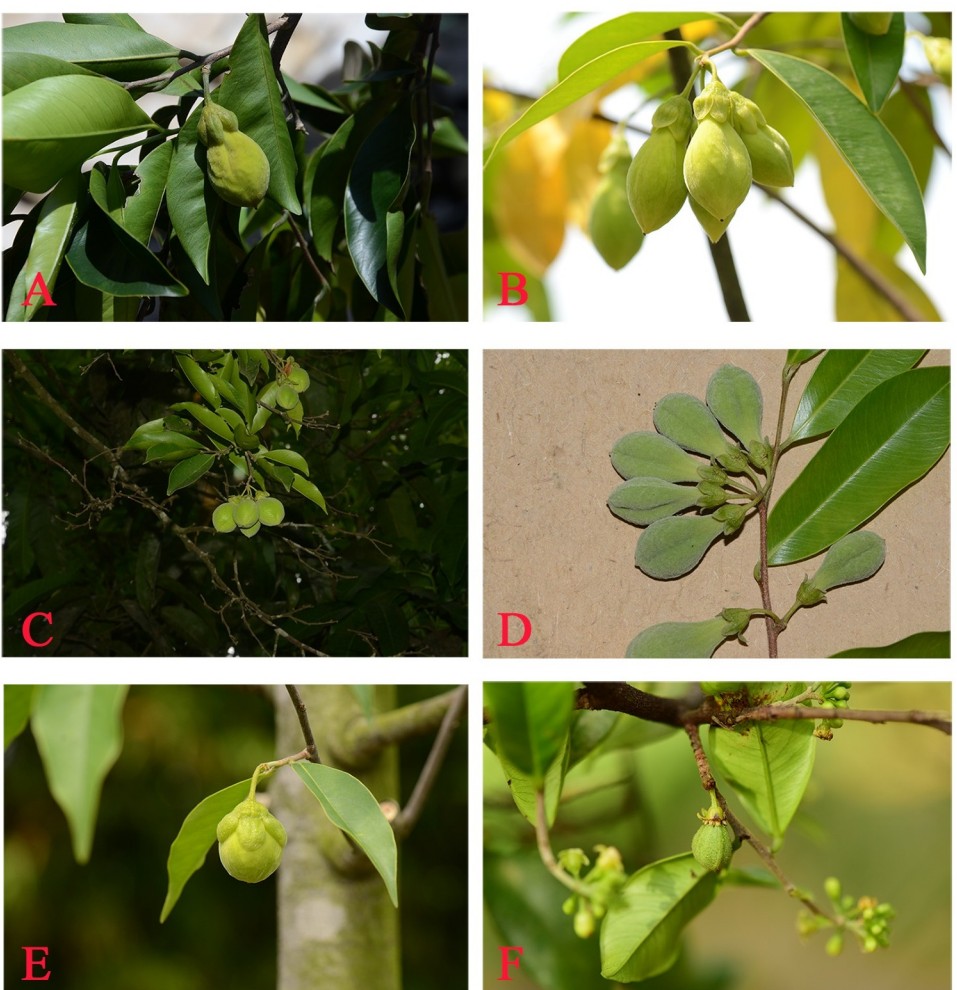

**Fig 2. Fruits of Qi-Nan germplasm and five accurately identified _Aquilaria_ species.** A: Qi-Nan germplasm. B: _A. sinensis_. C: _A. yunnanensis_. D: _A. malaccensis_. E: _A. crassna_. F: _A. hirta_.

**Table 1. Details on the PCR primers used in this study.**

| DNA barcode | Primer | Primer sequence (5'-3') | PCR conditions |
|---|---|---|---|
| ITS2 [21] | ITS-S2F | ATGCGATACTTGGTGTGAAT | 94˚C for 5 min; [94˚C for 30 s, 56˚C for 30 s, 72˚C for 45 s] × 40 cycles; 72˚C for 10 min. |
| | ITS-S3R | GACGCTTCTCCAGACTACAAT | |
| _matK_ (Kim, unpublished) | 3F_KIM | CGTACAGTACTTTTGTGTTTACGAG | 94˚C for 1 min; [94˚C for 30 s, 52˚C for 20 s, 72˚C for 50 s] × 35 cycles; 72˚C for 5 min. |
| | 1R_KIM | ACCCAGTCCATCTGGAAATCTTGGTTC | |
| _rbcL_ [22] | a_F | ATGTCACCACAAACAGAGACTAAAGC | 95˚C for 4 min; [94˚C for 30 s, 55˚C for 1 min, 72˚C for 1 min] × 35 cycles; 72˚C for 10 min. |
| | a_R | CTTCTGCTACAAATAAGAATCGATCTC | |
| _trnH-psbA_ [23, 24] | trnHf_05 | CGCGCATGGTGGATTCACAATCC | 94˚C for 5 min; [94˚C for 1 min, 55˚C for 1 min, 72˚C for 90 s] × 30 cycles; 72˚C for 7 min. |
| | psbA3-f | GTTATGCATGAACGTAATGCTC | |
| _trnL-trnF_ [25] | e | GGTTCAAGTCCCTCTATCCC | 94˚C for 5 min; [94˚C for 45 s, 50˚C for 45 s, 72˚C for 90 s] × 30 cycles; 72˚C for 10 min. |
| | f | ATTTGAACTGGTGACACGAG | |

success rate was evaluated according to the "best match", "best close match" and "all species barcodes" (BBA) method in TaxonDNA software [30] to identify the single fragment or combination with the highest success rate. Next, phylogenetic trees were generated using the Bayesian interference (BI) and maximum likelihood (ML) approaches in MrBayes 3.2.6 [31] and PAUP 4b (http://paup.phylosolutions.com), respectively. The clusters in the phylogenetic tree constructed by the best sequence combination were subsequently analyzed. Figtree 1.4.3 (http://tree.bio.ed.ac.uk/software/figtree/) were used generate visually appealing phylogenetic trees. Genetic distance and phylogenetic tree construction were mapped in R 4.0.0 (https://www.r-project.org). The GenBank accession numbers of all DNA fragments in this study are shown in S2 Table.

## Results

### PCR amplification and DNA sequencing

This test included 58 samples of Qi-Nan germplasm and 7 samples from *Aquilaria* species. The ITS2, *matK*, *trnH-psbA*, *rbcL* and *trnL-trnF* sequences of all samples were subjected to PCR amplification and sequencing. A total of 650 sequences were obtained by forward and reverse sequencing. The success of PCR amplification and sequencing, as well as the sequence length, variable sites, conserved sites, parsimony informative sites and singleton sites are shown in Table 2. PCR amplification of five DNA barcoding loci was successful in all samples. Except for the *trnL-trnF* sequence (which had a sequencing success rate of 0%), the other sequences achieved a sequencing success rate of 100%. Moreover, the sequencing quality of *trnL-trnF* was repetitive, which was not suitable for sequence alignment, assembly and analysis in this study. The number of variable sites for each sequence was as follows: ITS2 (11) > *matK* (9) > *rbcL* (1) = *trnH-psbA* (1). The number of conserved sites for each sequence was as

**Table 2. Evaluation of the five DNA barcode loci.**

| DNA barcode | PCR success (%) | Sequencing success (%) | Sequence length | No. of variable sites | No. of conserved sites | No. of parsimony informative sites | No. of singleton sites |
|---|---|---|---|---|---|---|---|
| ITS2 | 100 | 100 | 451 | 11 | 439 | 6 | 5 |
| *matK* | 100 | 100 | 682 | 9 | 672 | 6 | 3 |
| *rbcL* | 100 | 100 | 540 | 1 | 539 | 0 | 0 |
| *trnH-psbA* | 100 | 100 | 363 | 1 | 356 | 0 | 1 |
| *trnL-trnF* | 100 | 0 (Repetitive sequence) | - | - | - | - | - |
| ITS2+*matK* | - | - | 1133 | 20 | 1111 | 12 | 8 |
| ITS2+*rbcL* | - | - | 991 | 12 | 978 | 6 | 5 |
| ITS2+*trnH-psbA* | - | - | 814 | 12 | 795 | 6 | 6 |
| *matK*+*rbcL* | - | - | 1222 | 10 | 1211 | 6 | 3 |
| *matK*+*trnH-psbA* | - | - | 1045 | 10 | 1028 | 6 | 4 |
| *rbcL*+*trnH-psbA* | - | - | 903 | 2 | 895 | 0 | 1 |
| ITS2+*matK*+*rbcL* | - | - | 1673 | 21 | 1650 | 12 | 8 |
| ITS2+*matK*+*trnH-psbA* | - | - | 1496 | 21 | 1467 | 12 | 9 |
| ITS2+*rbcL*+*trnH-psbA* | - | - | 1354 | 13 | 1334 | 6 | 6 |
| *matK*+*rbcL*+*trnH-psbA* | - | - | 1585 | 11 | 1567 | 6 | 4 |
| ITS2+*matK*+*rbcL*+*trnH-psbA* | - | - | 2036 | 22 | 2006 | 12 | 9 |

follows: *matK* (672) > *rbcL* (539) > ITS2 (439) > *trnH-psbA* (356). The number of parsimony informative sites for each sequence was as follows: *matK* (6) = ITS2 (6) > *rbcL* (0) = *trnH-psbA* (0). Finally, the number of singleton sites for each sequence was as follows: ITS2 (5) > *matK* (3) > *trnH-psbA* (1) > *rbcL* (0). However, the fragment combinations with the largest number of variable sites were ITS2+*matK* (20), ITS2+*matK*+*rbcL* (21), ITS2+*matK*+*trnH-psbA* (21) and ITS2+*matK*+*rbcL*+*trnH-psbA* (22). It can also be inferred that ITS2 and *matK* play an important role in the genetic information sites of all primer fragments.

## Genetic distance

Of the four DNA barcodes, ITS2 and *matK* had large average genetic distances, while *trnH-psbA* had a smaller average genetic distance, the average genetic distance of *rbcL* was 0 (Fig 3). In the ITS2 region, the interspecific distance between the Qi-Nan germplasm and *A. crassna* was 0.0022±1.73E-18, the intraspecific distance among the Qi-Nan germplasms was 0, the interspecific distance between the Qi-Nan germplasm and *A. hirta* was 0.0203±2.08E-17, the interspecific distance between the Qi-Nan germplasm and *A. malaccensis* was 0.0157±0, the interspecific distance between the Qi-Nan germplasm and *A. sinensis* was 0, and the interspecific distance between the Qi-Nan germplasm and *A. yunnanensis* was 0.0090±1.04E-17. In the *matK* region, the interspecific distance between the Qi-Nan germplasm and *A. crassna* was 0.0091±5.20E-18, the intraspecific distance among the Qi-Nan germplasms was 0, the interspecific distance between the Qi-Nan germplasm and *A. hirta* was 0.0091±3.47E-18, the interspecific distance between the Qi-Nan germplasm and *A. malaccensis* was 0.0012±4.34E-19, the interspecific distance between the Qi-Nan germplasm and *A. sinensis* was 0, and the interspecific distance between the Qi-Nan germplasm and *A. yunnanensis* was 0.0039±3.90E-18. However, all genetic distances were 0 in the *rbcL* region. In the *trnH-psbA* region, the interspecific distance between the Qi-Nan germplasm and *A. hirta* was 0.0027±9.38E-07, the interspecific distance between the Qi-Nan germplasm and *A. sinensis* was 0.0027±9.47E-07, and the other genetic distances were 0. In addition, Wilcoxon signed-rank tests further confirmed that ITS2 and *matK* had the highest divergence in all genetic distances (Fig 3).

In the multilocus combinations, ITS2+*matK* had the highest genetic distances compared with the other barcode combinations (Fig 4). In the ITS2+*matK* region, the interspecific distance between the Qi-Nan germplasm and *A. crassna* was 0.0066±5.20E-18, the intraspecific distance among the Qi-Nan germplasms was 0, the interspecific distance between the Qi-Nan germplasm and *A. hirta* was 0.0132±1.73E-17, the interspecific distance between the Qi-Nan germplasm and *A. malaccensis* was 0.0066±4.34E-18, the interspecific distance between the Qi-Nan germplasm and *A. sinensis* was 0, and the interspecific distance between the Qi-Nan germplasm and *A. yunnanensis* was 0.0058±1.73E-18.

## Species discrimination

Preliminary evaluation of the DNA sequences showed that the *trnL-trnF* sequence was mostly repetitive and has a double peak. Therefore, only the 4 primers (ITS2, *matK*, *trnH-psbA* and *rbcL)* were selected for sequence screening and analysis. TaxonDNA analysis showed that the species identification success rate of each fragment or combination was different (Table 3). Of the single loci, ITS2 and *matK* were the best, the correct match rates of "best match", "best close match" and "all species" for these two sequences were 93.84%. In contrast, *rbcL* had the lowest successful identification rate (0.00%). The multi fragment combinations ITS2+*matK*, ITS2+*rbcL*, *matK*+*rbcL*, *matK*+*trnH-psbA*, ITS2+*matK*+*rbcL*, ITS2+*matK*+*trnH-psbA*, *matK*+*rbcL*+*trnH-psbA* and ITS2+*matK*+*trnH-psbA*+*rbcL* had the highest success rate, the correct match rates of "best match", "best close match" and "all species" for these combinations were

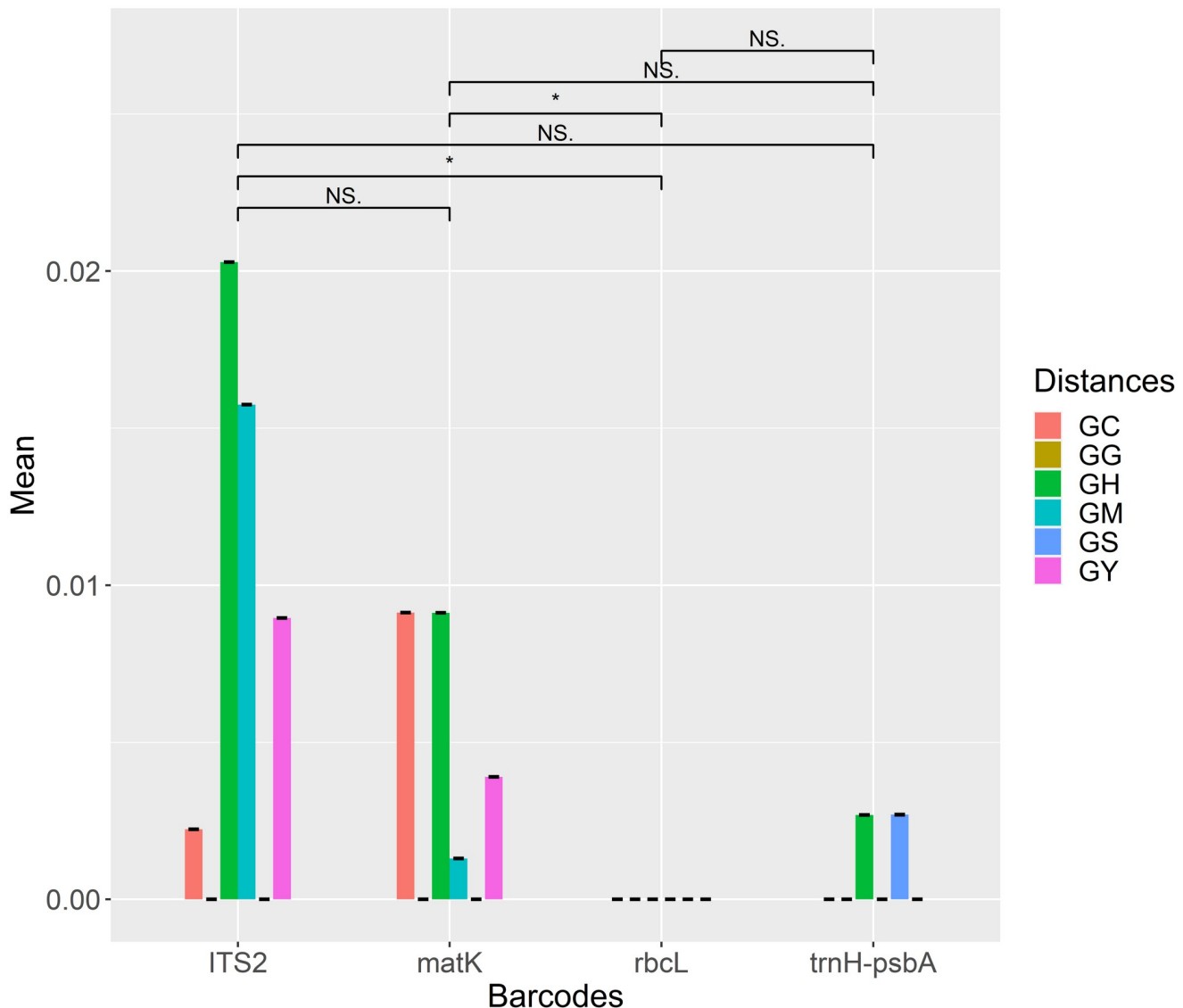

**Fig 3. Genetic distances between 58 Qi-Nan germplasms and the *Aquilaria* species for a single region.** (**GC**: interspecific distance between the Qi-Nan germplasm and *A. crassna*. **GG**: intraspecific distance among the Qi-Nan germplasms. **GH**: interspecific distance between the Qi-Nan germplasm and *A. hirta*. **GM**: interspecific distance between the Qi-Nan germplasm and *A. malaccensis*. **GS**: interspecific distance between the Qi-Nan germplasm and *A. sinensis*. **GY**: interspecific distance between the Qi-Nan germplasm and *A. yunnanensis*. *: P<0.05, NS: not significant).

93.84%. However, only ITS2+*matK* and ITS2+*matK*+*rbcL* had the lowest ambiguity (1.53%) under the "all species" method. In addition, the success rate of these two fragment combinations was equivalent to that of three or four of the other fragments. Therefore, to facilitate analysis, we selected ITS2+*matK* to construct the phylogenetic tree.

## Phylogenetic trees

The phylogenetic tree constructed with Bayesian inference (BI) and ITS2+*matK* is presented in Fig 5. The results showed that all 58 samples of Qi-Nan germplasm were significantly clustered with *A. sinensis_1*, *A. sinensis_2* and *A. sinensis_3*, but separate from *A. crassna*, *A. hirta*, *A. malaccensis* and *A. yunnanensis*. Additionally, the phylogenetic tree constructed with the

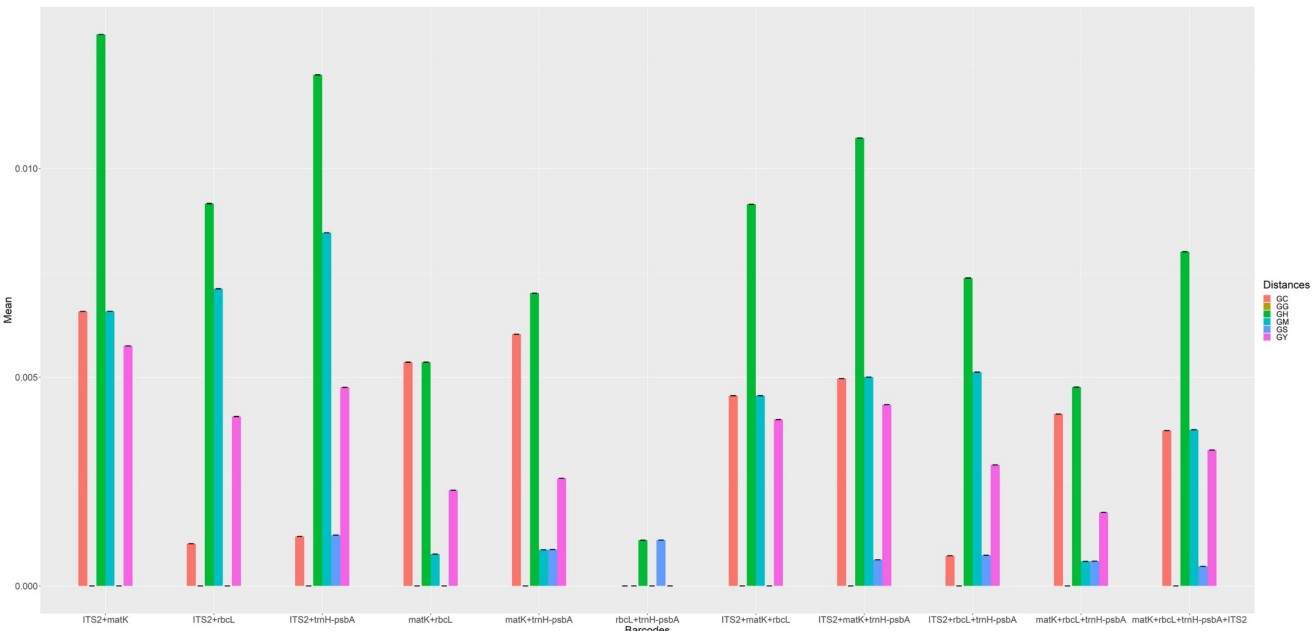

**Fig 4. Genetic distances between 58 Qi-Nan germplasms and the *Aquilaria* species for multilocus combinations.** (**GC**: interspecific distance between the Qi-Nan germplasm and *A. crassna*. **GG**: intraspecific distance among the Qi-Nan germplasms. **GH**: interspecific distance between the Qi-Nan germplasm and *A. hirta*. **GM**: interspecific distance between the Qi-Nan germplasm and *A. malaccensis*. **GS**: interspecific distance between the Qi-Nan germplasm and *A. sinensis*. **GY**: interspecific distance between the Qi-Nan germplasm and *A. yunnanensis*).

maximal likelihood method and ITS2+*matK* is presented in Fig 6. The results of ML analyses were similar to those of BI analyses: all Qi-Nan germplasms were clustered with *A. sinensis_1*, *A. sinensis_2* and *A. sinensis_3*. In conclusion, the phylogenetic tree clustering showed that the 58 Qi-Nan germplasms were genetic closest to *A. sinensis* but were less closely related to the other four *Aquilaria* species.

**Table 3. Species identification success rate based on TaxonDNA analysis.** C, A, I, and N represent correct, ambiguous, incorrect and no match, respectively.

| Region | Best match (%) | | | Best close match (%) | | | | All species (%) | | | |
|---|---|---|---|---|---|---|---|---|---|---|---|
| | C | A | I | C | A | I | N | C | A | I | N |
| ITS2 | 93.84 | 0.00 | 6.15 | 93.84 | 0.00 | 6.15 | 0.00 | 93.84 | 4.61 | 1.53 | 0.00 |
| *matK* | 93.84 | 0.00 | 6.15 | 93.84 | 0.00 | 6.15 | 0.00 | 93.84 | 3.07 | 3.07 | 0.00 |
| *rbcL* | 0.00 | 100.00 | 0.00 | 0.00 | 100.00 | 0.00 | 0.00 | 90.76 | 9.22 | 0.00 | 0.00 |
| *trnH-psbA* | 1.53 | 98.46 | 0.00 | 1.53 | 98.46 | 0.00 | 0.00 | 3.07 | 96.92 | 0.00 | 0.00 |
| ITS2+*matK* | 93.84 | 0.00 | 6.15 | 93.84 | 0.00 | 6.15 | 0.00 | 93.84 | 1.53 | 4.61 | 0.00 |
| ITS2+*rbcL* | 93.84 | 0.00 | 6.15 | 93.84 | 0.00 | 6.15 | 0.00 | 93.84 | 4.61 | 1.53 | 0.00 |
| ITS2+*trnH-psbA* | 93.84 | 0.00 | 6.15 | 93.84 | 0.00 | 6.15 | 0.00 | 4.61 | 93.84 | 1.53 | 0.00 |
| *matK*+*rbcL* | 93.84 | 0.00 | 6.15 | 93.84 | 0.00 | 6.15 | 0.00 | 93.84 | 3.07 | 3.07 | 0.00 |
| *matK*+*trnH-psbA* | 93.84 | 0.00 | 6.15 | 93.84 | 0.00 | 6.15 | 0.00 | 93.84 | 4.61 | 1.53 | 0.00 |
| *rbcL*+*trnH-psbA* | 1.53 | 96.92 | 1.53 | 1.53 | 96.92 | 1.53 | 0.00 | 3.07 | 96.92 | 0.00 | 0.00 |
| ITS2+*matK*+*rbcL* | 93.84 | 0.00 | 6.15 | 93.84 | 0.00 | 6.15 | 0.00 | 93.84 | 1.53 | 4.61 | 0.00 |
| ITS2+*matK*+*trnH-psbA* | 93.84 | 0.00 | 6.15 | 93.84 | 0.00 | 6.15 | 0.00 | 93.84 | 3.07 | 3.07 | 0.00 |
| ITS2+*rbcL*+*trnH-psbA* | 93.84 | 0.00 | 6.15 | 93.84 | 0.00 | 6.15 | 0.00 | 4.61 | 93.84 | 1.53 | 0.00 |
| *matK*+*rbcL*+*trnH-psbA* | 93.84 | 0.00 | 6.15 | 93.84 | 0.00 | 6.15 | 0.00 | 93.84 | 4.61 | 1.53 | 0.00 |
| ITS2+*matK*+*rbcL*+*trnH-psbA* | 93.84 | 0.00 | 6.15 | 93.84 | 0.00 | 6.15 | 0.00 | 93.84 | 3.07 | 3.07 | 0.00 |

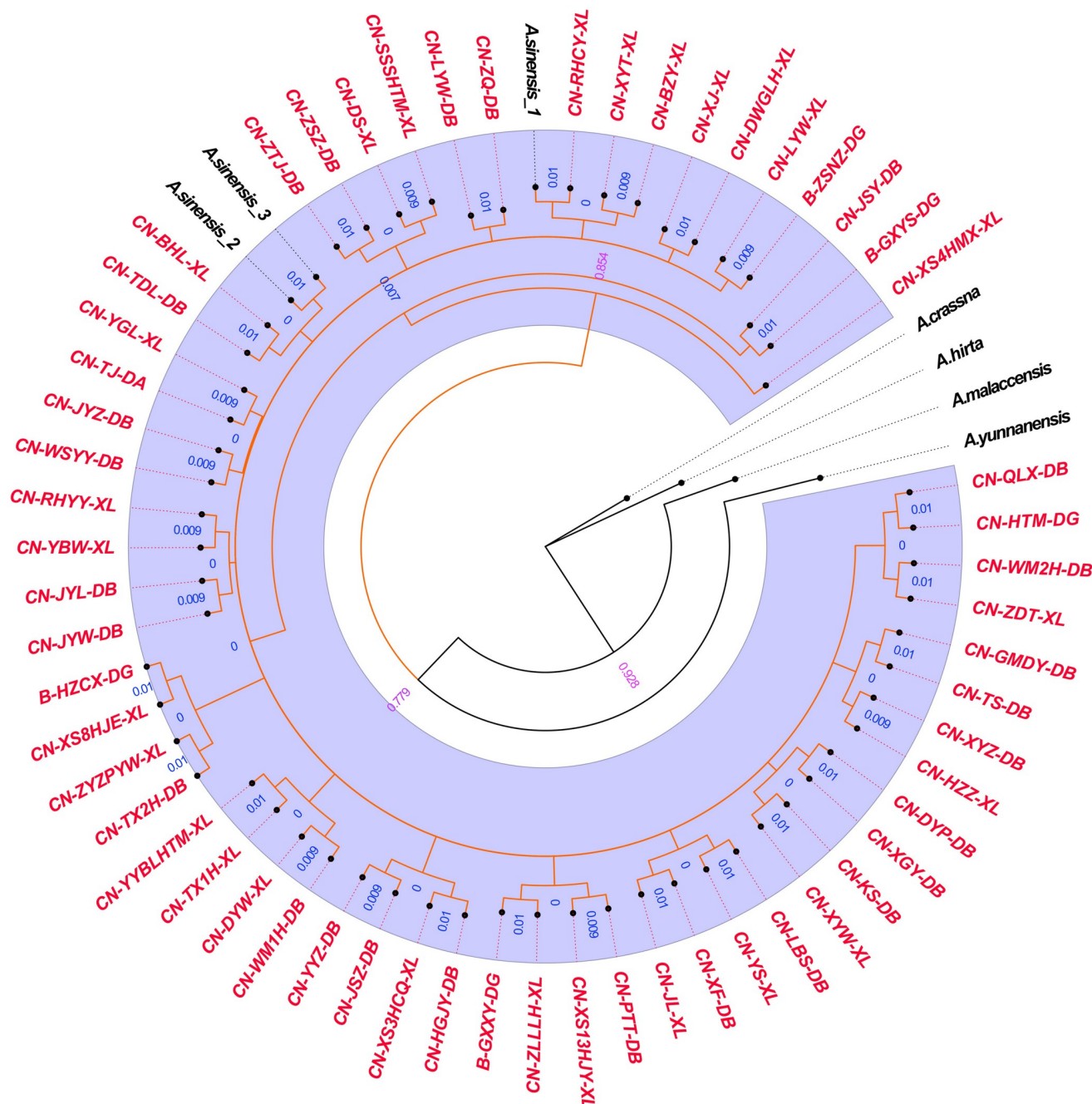

**Fig 5. The phylogenetic tree of the 58 Qi-Nan germplasms and five *Aquilaria* species, constructed with Bayesian inference and the ITS2 + *matK* combination.**

## Discussion

### DNA barcoding evaluation of the 58 Qi-Nan germplasms

According to the screening of the 5 DNA primers, single fragment ITS2 and *matK* had the most variation sites (Table 2). The phylogenetic tree clustering constructed by the ITS2+*matK*

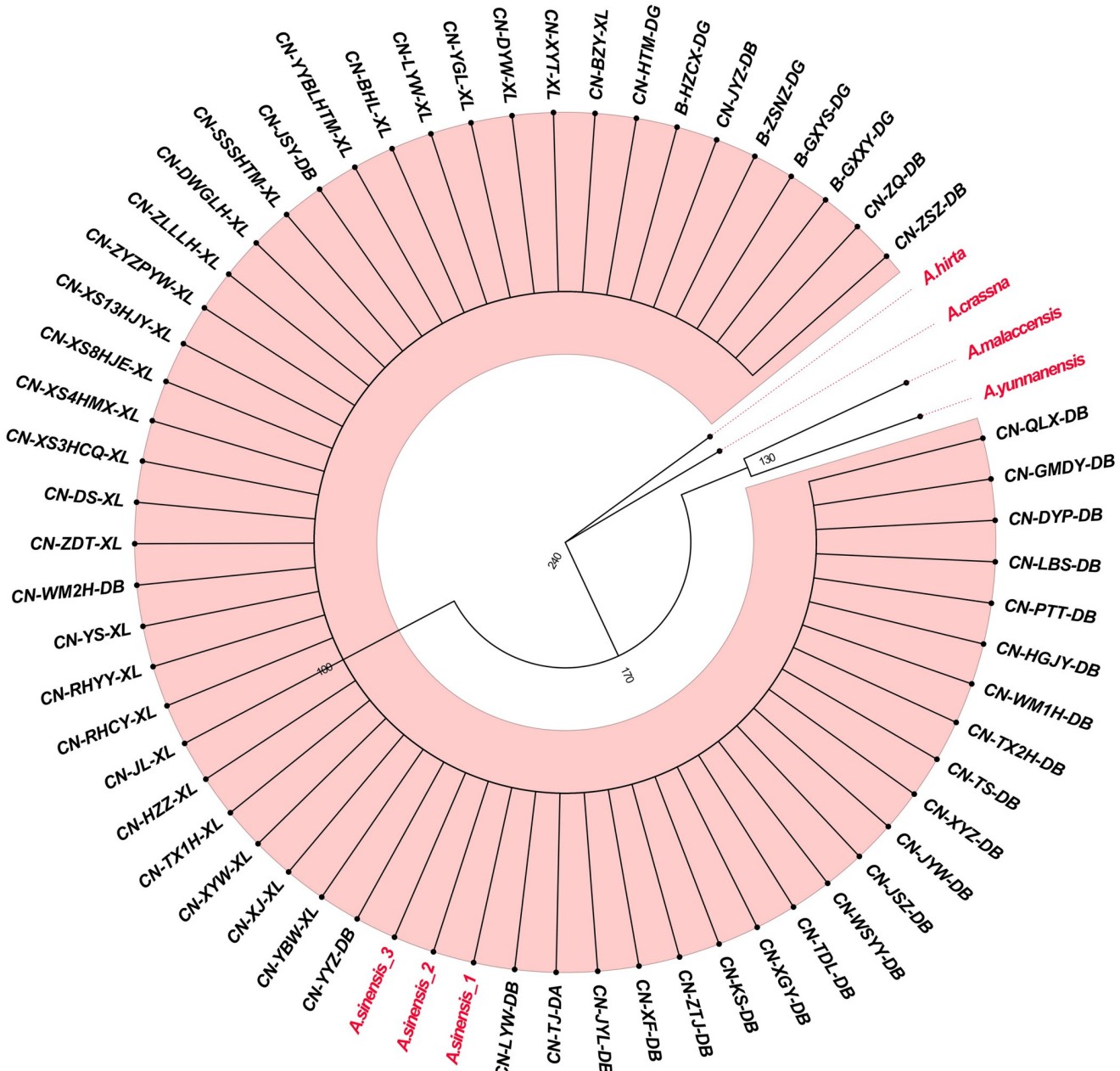

**Fig 6. The phylogenetic tree of the 58 Qi-Nan germplasms and five *Aquilaria* species, constructed with maximum likelihood and the ITS2 + *matK* combination.**

combination was the most significant. This finding was similar to the results of a preliminary analysis of *Aquilaria* species in different countries, which concluded that *matK* played an important role in identifying *Aquilaria* species [19]. Indeed, *matK* is one of the fastest-evolving genes in the protein-coding region of the chloroplast genome [32] and plays an important role in the molecular identification of many plants. For example, Steane et al. determined the evolution of *Casuarina* species with the *matK* sequence [33] and *Asparagus racemosus* is effectively identified with the *matK* sequence [34]. Furthermore, the *matK*+*rbcL* combination

helped identify *Acacia* accurately [35]. In addition, Chen et al. determined that the ITS2 sequence served as a universal barcode for medicinal plants based on extensive experimental data [36], as it has shorter elements than ITS and a higher success rate in PCR amplification and sequencing [37]. In addition, compared with cpDNA or nuclear barcodes alone, a combination of the two better identified different species [38].

The genetic distances between the 58 Qi-Nan germplasms and five *Aquilaria* species showed genetic divergences mainly in ITS2 and *matK*, while *trnH-psbA* had few divergences (Fig 3). In addition, the genetic distances of ITS2 and *matK* were largest between the Qi-Nan germplasm and *A. crassna*, *A. hirta*, *A. malaccensis*, and *A. yunnanensis*, but there was no genetic distance between the Qi-Nan germplasm and *A. sinensis*. Moreover, the genetic distances of *trnH-psbA* were largest between the Qi-Nan germplasm and *A. hirta* and *A. sinensis*, but these values were lower than 0.003. The *rbcL* fragment did not show any genetic distances between the Qi-Nan germplasm and *Aquilaria* species. Thus, we inferred that ITS2 and *matK* were ideal barcodes in this study [39], that variation in the *trnH-psbA* is low [40], and that the coding sequence of *rbcL* is highly conserved [41].

According to the species identification rates of 4 high quality sequences analyzed by the BBA method in TaxonDNA, the multifragment combinations ITS2+*matK*, ITS2+*rbcL*, *matK*+*rbcL*, *matK*+*trnH-psbA*, ITS2+*matK*+*rbcL*, ITS2+*matK*+*trnH-psbA*, *matK*+*rbcL*+*trnH-psbA* and ITS2+*matK*+*trnH-psbA*+*rbcL* had the highest success rate (Table 3). Since a 2-fragment combination is more efficient and reduces the cost of sequencing, the ITS2+*matK* combination was selected to analyze the clustering of the phylogenetic trees. This differs from previous research on the DNA barcoding identification of *Aquilaria* species, as Lee et al. concluded that the phylogenetic tree constructed by ITS2+*trnL-trnF* was suitable for *Aquilaria* species [5], and Li et al. found that the phylogenetic tree constructed by the combination of ITS+*matK* and ITS+*trnL-trnF* was conducive to species identification in three *Aquilaria* species [17]. Other research indicated that the *trnL-trnF* sequence could provide molecular identification of *Aquilaria* species [18]. However, the *trnL-trnF* sequence was not applicable in this study. This difference could possibly be explained by differences in test materials or tree-building methods with the adopted DNA barcodes or combinations. In the current study, the *trnL-trnF* sequence was mostly repetitive, which was not conducive to conducting a cluster analysis of the phylogenetic tree. Thus, ITS2+*matK* was selected for cluster analysis of the phylogenetic tree of Qi-Nan germplasm and *Aquilaria* species.

## Genetic relationship and source species of the 58 Qi-Nan germplasms

Through comparison of plant morphology, the fruit of the Qi-Nan germplasm was found to be the closest to the fruit of *A. sinensis* in shape and size (Fig 2). Previously, *Aquilaria* species were mainly classified by the characteristics of their flowers and fruits [5, 42, 43]. *A. sinensis* was chiefly identified by a moderate calyx that did not wrap the fruit, smooth seed coat without yellow pubescence, and long seed appendages. *A. yunnanensis* features oval fruit, a smaller and scattered calyx, and seeds densely coated by pubescence [6]. *A. malaccensis* has round fruit and a small calyx that degrades after the fruit ripens, and *A. crassna* features oval or relatively round fruit, a larger fruit and calyx, with the fruit usually wrapped in the calyx, and thick and leathery leaves [44]. Therefore, the source species of the Qi-Nan germplasm was inferred to be *A. sinensis* based on plant morphology.

Whether in the single regions or multilocus combinations, the intraspecific distance among the 58 Qi-Nan germplasms (GG) was 0, and the interspecific distance between the Qi-Nan germplasm and *A. sinensis* (GS) was the smallest (Figs 3 and 4). This finding indicates that different types of Qi-Nan germplasm significantly differ in plant morphology and agarwood

quality. However, the 58 Qi-Nan germplasms selected did not significantly differ in molecular identification, and all were most closely related to *A. sinensis*. Genetic distances can also reflect the relationship between different species and germplasms. For example, Zheng et al. found that ITS2 not only quickly and accurately identifies *Fritillaria cirrhosa* and its related species but also that the genetic relationship between different *Fritillaria* species is clearly explained by the genetic distance between *F. cirrhosa* and its related species [45]. When Zhang et al. analyzed the genetic distance and phylogenetic tree of *Phellodendron amurense* samples, they found that the genetic distance was important in the analysis and identification of the genetic relationship of *Phellodendron* species [46].

The BI and ML phylogenetic trees constructed by the ITS2+*matK* combination showed that all 58 Qi-Nan germplasms were closely related to *A. sinensis* but less closely related to *A. yunnanensis A. crassna*, *A. malaccensis* and *A. hirta* (Figs 5 and 6). Approximately 20 *Aquilaria* species are found in tropical parts of Southeast Asia [4, 5]. Huo et al. were one of the earliest to classify and describe the morphological characteristics of *Aquilaria* species, creating 12 categories of *Aquilaria* plants in the Thymelaeaceae family [44]. Recent research also indicated that *Aquilaria* plants in Asia could be divided into 13 species [47]. Of these, only *A. sinensis* and *A. yunnanensis* are found in China, the former is mostly distributed in Guangdong, Guangxi and Hainan Provinces, and the latter is only found in Xishuangbanna [6]. According to our preliminary visit to a plantation of Qi-Nan germplasm, the Qi-Nan germplasm currently cultivated in Guangdong was obtained by grafting (Fig 1). First, the branches of wild *Aquilaria* trees were grafted onto *A. sinensis*, after maturity the germplasm was propagated by grafting branches onto cultivated trees. The scion was mainly wild *A. sinensis* from Huizhou, Dianbai, Shenzhen, Hong Kong, and Hainan Province in China. According to the geographical distribution of *Aquilaria* species and the results of the phylogenetic tree, the source species of Qi-Nan germplasm cultivated in China is *A. sinensis*. However, DNA barcoding still has certain limitations and failed to resolve differences among Qi-Nan germplasm resources. Our group is currently attempting to carry out a thorough study using inter simple sequence repeats (ISSR) and random amplified polymorphic DNA (RAPD) molecular markers as well as other techniques. For example, ISSR and RAPD molecular marker techniques were used to analyze the genetic diversity of three widely planted Qi-Nan germplasms (A11, R21 and B31) in the market. It was found that they had high genetic diversity and high degree of genetic differentiation at the species level. And A11, R21 and B31 can be distinguished by genetic distance and cluster analysis [48, 49].

## The relationship between "Qi-Nan" agarwood and Qi-Nan germplasm

"Qi-Nan" agarwood has different names in different countries and regions, including Chinese names (e.g., Qinan, Jianan, and Jialuo) and English names (e.g., Qi-Nan, Kanankoh, Kyara and Chi-Nan) [50]. Historical records reported "Qi-Nan" agarwood as an *Aquilaria* species in the traditional sense, referring to top-grade agarwood formed under extremely demanding conditions that was rich in resin, elegant in fragrance and dark in color. These harsh conditions mainly mean that the incense tree is damaged by lightning and thunder, hurricane blowing, insect bites or artificial felling, and the formation of agarwood takes decades or even hundreds of years [51]. It was named for its mysterious scent that could be achieved without burning the wood and was distinguished from other types of agarwood as the most expensive and top-quality due to its unique smell and appearance [52]. "Qi-Nan" agarwood is further divided according to appearance and color into green Qi-Nan, purple Qi-Nan, black Qi-Nan, yellow Qi-Nan, etc. [51, 53]. At present, the market price of "Qi-Nan" agarwood has far exceeded that of general agarwood.

## Conclusion

We showed in this study that a combination barcode of ITS2+*matK* is useful for source species identification of 58 Qi-Nan germplasms. This paper is the first to use DNA barcoding to identify Qi-Nan germplasm cultivated in China and report that it originated from *A. sinensis.* These findings may inform the future promotion and application of agarwood produced from Qi-Nan germplasm. First of all, this will help us to master the growth habit and morphological characteristics of different varieties of Qi-Nan germplasm, as well as the selection and breeding of new varieties of *Aquilaria* species in the future by carrying out the investigation of Qi-Nan germplasm. Secondly, the use of DNA barcoding to identify the source species of different varieties of Qi-Nan germplasm can build a molecular identification system of Qi-Nan germplasm, which is helpful to analysis the relationship of Qi-Nan germplasm with domestic and foreign *Aquilaria* species. Finally, we can formulate classification standards for the use of this germplasm, and provide identification information of different varieties of Qi-Nan germplasm in the resource market, scientific research and authenticity identification of agarwood in the future.

## Supporting information

**S1 Fig. Whole plants of selected Qi-Nan germplasms.** A: CN-TJ-DA. B: CN-QLX-DB. C: CN-JSY-DB. D: CN-DYP-DB. E: CN-LBS-DB. F: CN-LYW-DB. G: CN-ZTJ-DB. H: CN-ZSZ-DB. I: CN-XS3HCQ-XL.
(TIF)

**S1 Table. List of the samples of 58 Qi-Nan germplasms and five *Aquilaria* species used in this study.** Includes the name, sample numbers, origins, locations and notes of each sample.
(XLSX)

**S2 Table. GenBank accession numbers of the Qi-Nan germplasm and *Aquilaria* species generated in this study.**
(XLSX)

## Acknowledgments

We are very grateful for the enthusiastic help of the local cultivators in collecting samples from the Qi-Nan germplasm plantations in Guangdong and Hainan Provinces in China.

## Author Contributions

**Conceptualization:** Yun Yang, Jianhe Wei.

**Data curation:** Yong Kang.

**Formal analysis:** Yong Kang.

**Investigation:** Yong Kang, Peiwei Liu, Feifei Lv, Yuxiu Zhang, Yun Yang.

**Project administration:** Jianhe Wei.

**Supervision:** Jianhe Wei.

**Visualization:** Yong Kang.

**Writing – original draft:** Yong Kang.

**Writing – review & editing:** Yong Kang, Yun Yang, Jianhe Wei.

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
