## [Decision Letter · Decision Letter 0]

7 Apr 2022

PONE-D-22-07044Genetic relationship and source species identification of 58 Qi-Nan germplasms of Aquilaria species in China that easily form agarwoodPLOS ONE

Dear Dr. KANG,

Thank you for submitting your manuscript to PLOS ONE. After careful consideration, we feel that it has merit but does not fully meet PLOS ONE’s publication criteria as it currently stands. Therefore, we invite you to submit a revised version of the manuscript that addresses the points raised during the review process.

We look forward to receiving your revised manuscript.

Kind regards,

Pankaj Bhardwaj, Ph.D.

Academic Editor

PLOS ONE

Journal Requirements:

2. Please remove your figures from within your manuscript file, leaving only the individual TIFF/EPS image files, uploaded separately.  These will be automatically included in the reviewers’ PDF.

Reviewers' comments:

Reviewer's Responses to Questions

**Comments to the Author**

1. Is the manuscript technically sound, and do the data support the conclusions?

Reviewer #1: Yes

Reviewer #2: Yes

2. Has the statistical analysis been performed appropriately and rigorously? 

Reviewer #1: Yes

Reviewer #2: Yes

3. Have the authors made all data underlying the findings in their manuscript fully available?

Reviewer #1: Yes

Reviewer #2: Yes

4. Is the manuscript presented in an intelligible fashion and written in standard English?

Reviewer #1: Yes

Reviewer #2: Yes

5. Review Comments to the Author

Reviewer #1: The authors have attempted to identify germplasm resources of Aquilaria species using DNA barcoding analysis with the help of one nuclear and chloroplast genes or their combination. The manuscript is well written and the results are clearly presented. However, to get a better resolution of genetic differences, it is highly recommended to use molecular markers like SSR or ISSR or a combination of DNA barcoding and these molecular markers.

Reviewer #2: “Qi-Nan” is an important agarwood species valued for its fragrance and medicinal importance. Authors have made efforts to identify its source species using DNA barcoding analysis and inferred A. sinensis. to be the source species of “Qi-Nan”. The overall quality of manuscript the way it is written needs to be improved. My major concerns are as below:

1. Please mark Figure 1 properly

2. In line 136, please provide the full detail of the DNA isolation kit used.

3. It will be appropriate the address the primer pairs used in this study as universal primers rather than common primers (LINE 136)

4. Please rewrite methods sections, some of the information is duplicated while some sections need elaboration e.q.1) Data analysis section contain the duplicated information

2) the name of sequencing platform and the methodology of sequencing is not provided

5. Reference 2 is not complete

6. In general, few references are misleading in the manuscript. Authors are requested to recheck the citation used in the manuscript

7. It would have been appropriate to dedicate the last section of discussions to conclusion and future aspects but the last section appears more of the introductory part of the manuscript

6. PLOS authors have the option to publish the peer review history of their article (what does this mean?). If published, this will include your full peer review and any attached files.

Reviewer #1: No

Reviewer #2: No

---

## [Author Response · Author response to Decision Letter 0]

12 May 2022

Dear Dr. Pankaj Bhardwaj,

We would like to thank you and the two reviewers for your careful and thoughtful consideration of our manuscript. In particular, we were pleased that the Editor and reviewers were interested in the identification of Qi-Nan germplasms. We have taken care to integrate the suggestions brought up during review and feel that the paper is greatly improved by the process. We are confident that we have addressed the concerns brought up during review.

Sincerely,

Yong Kang

Comment is from Reviewer 1

Comment: The authors have attempted to identify germplasm resources of Aquilaria species using DNA barcoding analysis with the help of one nuclear and chloroplast genes or their combination. The manuscript is well written and the results are clearly presented. However, to get a better resolution of genetic differences, it is highly recommended to use molecular markers like SSR or ISSR or a combination of DNA barcoding and these molecular markers.

Response: Thank you very much for your advice. DNA barcoding technology has been widely used in species identification, especially in the identification of closely related species and germplasm resources. It has the advantages of strong objectivity, high accuracy and rapid identification. The combinations ITS2+matK had the highest success rate, the correct match rates of “best match”, “best close match” and “all species” for these combinations were 93.84% in this study. The results can support the accuracy of identification of the source species of Qi-Nan germplasms. However, I will focus on the combination of SSR or ISSR with DNA barcoding to get a better resolution of genetic differences in the future research.

Comments are from Reviewer 2

“Qi-Nan” is an important agarwood species valued for its fragrance and medicinal importance. Authors have made efforts to identify its source species using DNA barcoding analysis and inferred A. sinensis. to be the source species of “Qi-Nan”. The overall quality of manuscript the way it is written needs to be improved.

The major concerns of reviewers are as below:

Comment 1: Please mark Figure 1 properly.

Response: Thanks for your suggestion. Figure 1 has been marked.

Comment 2: In line 136, please provide the full detail of the DNA isolation kit used.

Response: Thank you for your comments. Full details of the DNA isolation kit used have been supplemented.

Comment 3: It will be appropriate the address the primer pairs used in this study as universal primers rather than common primers (LINE 136).

Response: Thanks very much. I have finished the modification according to your comments.

Comment 4: Please rewrite methods sections, some of the information is duplicated while some sections need elaboration e.q.1) Data analysis section contain the duplicated information. 2) the name of sequencing platform and the methodology of sequencing is not provided.

Response: Thank you very much for your review. I have rewritten the method section. On the one hand, the content of duplicate information in data analysis has been modified. And on the other hand, DNA sequencing methods and platforms have been supplemented.

Comment 5: Reference 2 is not complete.

Response: Thanks for your suggestion. I have replaced reference 2.

Comment 6: In general, few references are misleading in the manuscript. Authors are requested to recheck the citation used in the manuscript.

Response: Thank you for your valuable comments. I have rechecked and revised the citations used in the manuscript.

Comment 7: It would have been appropriate to dedicate the last section of discussions to conclusion and future aspects but the last section appears more of the introductory part of the manuscript.

Response: Thank you very much for your advice. I have revised in the discussion and conclusion according to your comments.

---

## [Decision Letter · Decision Letter 1]

6 Jun 2022

Genetic relationship and source species identification of 58 Qi-Nan germplasms of Aquilaria species in China that easily form agarwood

PONE-D-22-07044R1

Dear Dr. KANG,

We’re pleased to inform you that your manuscript has been judged scientifically suitable for publication and will be formally accepted for publication once it meets all outstanding technical requirements.

Kind regards,

Pankaj Bhardwaj, Ph.D.

Academic Editor

PLOS ONE

Reviewers' comments:

Reviewer's Responses to Questions

**Comments to the Author**

1. If the authors have adequately addressed your comments raised in a previous round of review and you feel that this manuscript is now acceptable for publication, you may indicate that here to bypass the “Comments to the Author” section, enter your conflict of interest statement in the “Confidential to Editor” section, and submit your "Accept" recommendation.

Reviewer #1: All comments have been addressed

Reviewer #2: All comments have been addressed

2. Is the manuscript technically sound, and do the data support the conclusions?

Reviewer #1: Yes

Reviewer #2: Yes

3. Has the statistical analysis been performed appropriately and rigorously? 

Reviewer #1: Yes

Reviewer #2: Yes

4. Have the authors made all data underlying the findings in their manuscript fully available?

Reviewer #1: Yes

Reviewer #2: Yes

5. Is the manuscript presented in an intelligible fashion and written in standard English?

Reviewer #1: Yes

Reviewer #2: Yes

6. Review Comments to the Author

Reviewer #1: (No Response)

Reviewer #2: Authors have made attempts to identify Qi-Nan germplasm using DNA barcoding. The combination of barcodes identified in the present study will facilitate the breeding programs and agarwood production in future.

Authors have incorporated the suggestions. I have made few correction in the annotated manuscript. Authors are requested to carefully check the manuscript for minor typographical errors.

7. PLOS authors have the option to publish the peer review history of their article (what does this mean?). If published, this will include your full peer review and any attached files.

Reviewer #1: No

Reviewer #2: No

---

## [Editor Report · Acceptance letter]

9 Jun 2022

PONE-D-22-07044R1 

Genetic relationship and source species identification of 58 Qi-Nan germplasms of *Aquilaria* species in China that easily form agarwood 

Dear Dr. Kang:

I'm pleased to inform you that your manuscript has been deemed suitable for publication in PLOS ONE. Congratulations! Your manuscript is now with our production department. 

Kind regards, 

on behalf of

Dr. Pankaj Bhardwaj 

Academic Editor

PLOS ONE